

# Technical note: „U-Th Analysis" – an open-source software dedicated to MCICPMS U-series-data treatment and evaluation

Inga Kristina Kerber[1], Fabian Kontor[1], Sophie Warken[1,2], Norbert Frank[1] *

[1] Institute for Environmental Physics, Heidelberg University, Heidelberg, Germany
[2] Institute for Earth Sciences, Heidelberg University, Heidelberg, Germany

*Correspondence to*: Norbert Frank (mailto:norbert.frank@iup.uni-heidelberg.de)

## Abstract

We present our standalone data analysis application for $^{230}$Th/U dating on multi-collector inductively coupled plasma mass spectrometers (MC-ICP-MS). The Python-based algorithm is equipped with a graphical user interface (GUI) and comprises raw data treatment, corrections, age calculation, and error estimation. Our underlying measurement protocol employs a combination of Faraday cups (FC) and secondary electron multipliers (SEM), and the software allows for different detector layouts for the measurement of the least abundant isotopes $^{234}$U, $^{230}$Th and $^{229}$Th. We especially focus on features that ensure reproducibility and enable user-friendly reanalysis of measurements such as customized calculation constants with templates. Result files are saved automatically and contain all relevant settings used. Eventually, we demonstrate the relevance of adequate data outlier treatment and generally recommend using the median instead of the mean of calculated ratios. The performance of our evaluation software is demonstrated by a case study from a Puerto Rican stalagmite with growth phases from modern to 40 ka old. The majority of the obtained ages reaches uncertainties in the range of 0.3-0.6%, underlining the capability of our measurement protocol.

## 1 Introduction

The U-series disequilibrium method, $^{230}$Th/U dating, is a precise chronometer covering approximately the last 650 kiloyears, and has proven indispensable for the age determination of marine and continental carbonate archives and their applications (Bourdon et al., 2003). The method is based on a complete disequilibrium of $^{234}$U, with its daughter nuclide $^{230}$Th, during the formation of secondary carbonates. It presumes a subsequent closed system evolution of the activity ratio of ($^{230}$Th/$^{234}$U) and ($^{234}$U/$^{238}$U) since the time of formation. Ideally, the initial $^{230}$Th activity of the material is presumed zero or can be estimated from the total Th concentration via an initial ($^{230}$Th/$^{232}$Th) activity ratio. The dating applications for secondary carbonates and other appropriate materials are manifold in geochemistry, archaeology, and climate science. The applications relying on this dating method are manifold in geochemistry, archaeology and climate science. Further development of this dating method includes both improvements in instrumentation and measurement protocols, as well as reproducible data analysis and age calculation schemes (Pourmand et al., 2014;Andersen et al., 2004;Cheng et al., 2013;Breton et al., 2015;Chiang et al., 2019;Hellstrom, 2003;Hoffmann et al., 2007;Shen et al., 2002;Shen et al., 2012;Kerber et al., 2023;Shao et al., 2019). The presently most sensitive and precise technology for high precision U and Th isotope measurements is multi-collector inductively coupled plasma source mass spectrometry (MC-ICPMS). Recent technological



advances of MC-ICP-MS include the implementation of high ohmic amplifiers allowing to enhance the dynamic
range of multiple Faraday-collectors (FC) to six orders of magnitude for the simultaneous detection of very large
and low isotope abundances , instead of the conventionally used combination of secondary electron multipliers
(SEM) and FC(Breton et al., 2015). Measurement protocol updates aim at increasing measurement precision
and/or decreasing input sample masses by combining new detector layouts, improving the understanding of
correction factors, and ensuring a stable measurement environment (Cheng et al., 2013;Chiang et al., 2019;Shen
et al., 2002;Shen et al., 2012;Hellstrom, 2003;Andersen et al., 2004;Hoffmann et al., 2007;Kerber et al.,
2023;Shao et al., 2019).
We here focus on the third route for the enhancement of $^{230}$Th/U dating, which is clear and reproducible data
analysis and age calculation schemes. Up to now, only two $^{230}$Th/U dating data analysis routines have been
published (Shao et al., 2019;Pourmand et al., 2014). However, regarding the rising amount of data being produced
in MC-ICP-MS laboratories, data management is becoming more and more important. For example, some samples
might require later adaptation of the individual corrections of isotope ratios due to residual contamination with
non-carbonate material or detection of initial $^{230}$Th from the carbonate forming environment.
Dating young materials of only a few years to centuries in age is challenging (i) due to the small number of counts
on especially $^{230}$Th, which implies that all correction factors including "ghost signal" corrections need to be
determined very precisely (Zhao et al., 2009;Chiang et al., 2019;Kerber et al., 2023). Regarding the removal of
scatter ions on the specific low abundance masses 230 and 229 amu, Kerber et al. (2023) demonstrated an effective
correction based on a linear dependence of the scattered ions on the $^{238}$U signal. Other authors separate U and Th
chemically to reduce or remove the $^{238}$U beam from the low abundance Thorium isotope measurements (Chiang
et al., 2019), which implies flexibility in the detector arrangement and data treatment protocol. As such scatter
peaks may depend on the specific instrument or vary through time, these corrections need to be adaptable constants
in the data evaluation routine. The influence on final atomic ratio and accuracy of the ghost signals as well as by
typical variation in other individual corrections such as peak tailing, mass fractionation, isobaric interferences are
evaluated in detail in Kerber et al. (2023).
In addition, the correction for initial Thorium may cause large age corrections and propagated uncertainties, in
particular since adequate initial Th values based on the $^{230}$Th/$^{232}$Th ratio may be variable and difficult to detect
(Hellstrom, 2006;Wenz et al., 2016;Wortham et al., 2022). There are different methods to estimate the initial Th
isotope ratio: First, isochrons can be used to determine the isotopic composition of the detrital component in the
carbonate (Ludwig and Titterington, 1994;Wenz et al., 2016;Stinnesbeck et al., 2020;Töchterle et al., 2022).
Secondly, analyses of modern drip waters or recent carbonate deposits allow estimation of the value and sources
of initial Th (Wortham et al., 2022;Li et al., 2022). In some cases the "true" age of a stalagmite can be also inferred
from other dating methods, such as radiocarbon (Akers et al., 2019;Huang et al., 2024) or the stratigraphic order
(Hellstrom, 2006). Also, several approaches can be combined (Warken et al., 2020;Akers et al., 2016;Roy-Barman
and Pons-Branchu, 2016).
Other aspects are updating half-lives (such as e. g. from Cheng et al. (2000b) to Cheng et al. (2013)), which makes
re-evaluation of previously measured data necessary. These tasks are error-prone, in particular when they require
copy-and-pasting data in e. g. spreadsheets. Also, a clear and unified documentation of the applied constants and
the way of saving data is desirable. Additionally, the statistical methods, for example for outlier correction, should



undergo clear documentation. Altogether, this helps to report Th/U ages in a standardized way (Dutton et al.,
77    2017).

In this study, we present our user-friendly GUI and the underlying algorithm for data treatment and age
calculation. Methods to treat outliers in measurement data are particularly highlighted. As a case study, we present
newly obtained ages from a stalagmite from Larga Cave, Puerto Rico, which shows a modern growth phase, as
well as continuous deposition during the last Glacial into the deglaciation, thus demonstrating the performance of
our method for both very young and older sample materials. Our protocol enables a precise determination of
speleothem growth rates, which allows a comparison to a coevally deposited stalagmite from the same cave
highlighting the influence of in-cave processes on speleothem growth rates. In particular, this dataset showcases
how initial $^{230}$Th correction models can be easily tested with our here presented software and GUI, and how those
influence speleothem chronologies.

## 2    Methods

### 2.1    Standards and reference materials

We use our in-house triple spike solution (TriSpike) with a $^{233}$U concentration of (0.038556 ± 0.0000009) ng/g, a
$^{236}$U concentration of (3.86778 ± 0.00009) ng/g and a $^{229}$Th concentration of (0.018055 ± 0.000008) ng/g (2
standard error of the mean) (Kerber et al., 2023). For standard bracketing, we employ the Harwell-Uraninite 1
(HU-1) as a reference material. Its activity ratios ($^{230}$Th/$^{238}$U) and ($^{234}$U/$^{238}$U) are presumed to be 1, as it is a secular
equilibrium material. Abundance sensitivity and hydride correction are determined by measuring CRM-112A U
reference solution and an in-house $^{232}$Th standard. The CRM-112A gravimetric standard solution has a $^{238}$U
concentration of (4.3021 ± 0.0015) μg/g, while the inhouse $^{232}$Th standard calibrated with TriSpike has a $^{232}$Th
concentration of (505.8 ± 1.02) ng/g (2 σ uncertainties). CRM-112A solution is also used to track the values of
the two ghost signal constants, $k_{229}$ and $k_{230}$ (Kerber et al., 2023). For $k_{229}$, it is measured without addition of
TriSpike, while in the case of $k_{230}$, the spiked CRM-112A solution is employed. For age determination, the $^{230}$Th
and $^{234}$U decay constants determined by Cheng et al. (2013) are used. Ages are reported with 2 σ statistical standard
mean error, but  do not include half-life uncertainties.

### 2.2    Chemical preparation and instrumentation

The chemical preparation of carbonate samples includes sample dissolution in ultra clean nitric acid, spiking with
TriSpike and two steps of wet column chromatographic ion exchange separation of U and Th from matrix elements
using Eichrom UTEVA resin (Douville et al., 2010;Wefing et al., 2017;Matos et al., 2015). Chemical blanks are
commonly below 0.4 fg for $^{234}$U and 0.04 fg for $^{230}$Th and Ca matrix concentrations are required to be below 10
ppm. For the mass-spectrometric measurement, samples are dissolved in 1 % HNO$_3$ and 0.05 % HF. All samples
were measured on a MC-ICP-MS (ThermoFisher Neptune Plus) at the Institute for Environmental Physics,
Heidelberg University (Germany). The mass spectrometer is equipped with Faraday cups (FC) and a central
secondary electron multiplier (SEM). The central detector can be selected between the SEM and a FC connected
to a $10^{13}$ Ω amplifier. $^{238}$U is measured on a $10^{10}$ Ω amplified resistor. All other FC are connected to $10^{11}$ Ω
amplifiers. The desolvating system CETAC Aridus II is used as inlet. A measurement sequence starts with the
determination of abundance sensitivity and tailing on two different solutions for both uranium and thorium. Each
sample and standard measurement is preceded by a procedural blank measurement to ensure that the background



signal has gone back to a clean state. CRM-112A measurements are carried out to track ghost signal values at the
beginning and end of a measurement sequence. Samples are bracketed with HU-1 as a reference material. The
GUI is written for this type of measurement protocol. Other adaptions such as fewer procedural blank
measurements or else require small changes in the code, but are easily feasible.

### 2.3    Speleothem sample description

Stalagmite B1 was collected in 2019 in Larga cave, Puerto Rico (18°19′N 66°48′W, 350msl, supplementary Figure
S1A) from a passage in the deep part of the cave connected to the "Collapse room". The host rock overburden at
the location of the sample is about 40-60m. It is in total 60 cm long, and has an average diameter of 15 cm
(supplementary Figure S1B). The drip site was still active, and was monitored with spot measurements over
several years, revealing varying drip intervals between 2 s and >120 s. A total amount of 37.7 ml water from the
drip site of stalagmite B1 was analysed for its U and Th activity ratios. Samples for $^{230}$Th/U dating of the
speleothem with typical input masses of 100-150 mg have been cut using a diamond wire saw along the growth
axis. Chemical preparation, mass-spectrometric measurements, data treatment and evaluation of drip water and
the speleothem samples followed the methods described in Kerber et al. (2023) and in this study.
Larga Cave is located in the north central karst region of Puerto Rico (supplementary Figure S1A). Previous work
including extensive cave air and drip monitoring has demonstrated that the cave is a valuable location to study of
the influence of changing climate on past rainfall patterns in the Western tropical Atlantic (Vieten et al.,
2018b;Warken et al., 2022a;Vieten et al., 2018a). In particular, the main passage of Larga Cave is subject to a
seasonally varying ventilation, which results in pCO$_2$ values of 600 ppm close to atmospheric values during
winter, and higher values up to 1800 ppm in summer (Vieten et al., 2016). In contrast, in the deep part of the cave,
where also stalagmite B1 was collected, ventilation is strongly muted, and cave air pCO$_2$ values are higher with
values up to 2300 – 3600 ppm (Vieten et al., 2016). As a result of this ventilation regime, growth rates are expected
to vary both seasonally, but also between different locations inside the cave (Vieten and Hernandez, 2021).
So far, two speleothem records from Larga Cave are been published, where the most recent covers the past 500
years (Vieten et al., 2024), and the second stalagmite grew during the period of 46.2-15.3 ka with a hiatus from
41.1 to 35.5 ka (Warken et al., 2020). For $^{230}$Th/U dating of Larga speleothems, high initial Th contents have to
be considered - a phenomenon that regularly occurs in speleothem records from the Caribbean and Central
American region (Fensterer et al., 2010;Steidle et al., 2021;Moseley et al., 2015;Schorndorf et al.,
2023;Stinnesbeck et al., 2020;Beck et al., 2001;Akers et al., 2016;Rivera-Collazo et al., 2015).

### 3    Data treatment and analysis procedures

The whole analysis procedure from raw data treatment to age calculation is conducted in one GUI featuring three
tabs: 'Input' for isotopic ratio calculations, 'Inspect' for outlier correction of the signal and 'Analysis' for age
calculation. The source code is accessible at https://github.com/EnvArchivesHD/UTh_Analysis. It is based on
the  open  source  PyQt5  Python  library  (https://pypi.org/project/PyQt5/).  The  folder
https://github.com/EnvArchivesHD/UTh_Analysis/tree/main/dist contains the compiled .exe file for the GUI
("UTh Data Analysis.exe") as well as default configuration files ("constants – coral.cfg" and "constants –
stalag.cfg". Input and output format of files are .csv or .xlsx. The GUI consists of three consecutive tabs, for which
the functionalities and the underlying calculations and processes will be described in the following.



### 3.1 Input tab

In 'Input', as presented in Figure 1, the user can navigate to the folder containing the raw mass spectrometer data and start the calculation of corrected isotopic ratios (Box 1 in Figure 1). All tab screenshots present data from stalagmite B1. Prior to the calculations, a configuration file containing all necessary constants used in the calculations needs to be loaded (same Box 1). This file contains constants and correction factors used for evaluation of the activity ratios and ages, such as mass fractionation coefficients, decay constants, the exact masses of the isotopes and the values applied for initial $^{230}$Th correction model. All constants can be edited manually either in the configuration file directly, or within the GUI using the button "edit". An exemplary configuration table is also provided in the supplementary material (Figure S2). To apply a $^{230}$Th correction model a value can be set for the activity ratio and uncertainty of the contaminating material ("A230Th232Th Init. "). The standard value would be the bulk earth mean activity ratio of $0.75 \pm 0.38$. Exemplary templates for corals and speleothems with conventionally used correction models are provided. For speleothems, a typical activity ratio of $(^{230}\text{Th}/^{232}\text{Th})_{\text{ini/detr}}$ of detritus is estimated to $0.75 \pm 0.38$, which is derived from a bulk earth Th/U weight ratio of $4.1 \pm 2.05$ (Wedepohl, 1995) and the assumption of $^{230}$Th, $^{234}$U, and $^{238}$U being in secular equilibrium (Cheng et al., 2013). Nevertheless, this ratio may require adjustment according to local conditions. The coral template assumes as default value an activity ratio of $8 \pm 4$, which is estimated for corals dwelling in waters of the northeast Atlantic upper thermocline (Wefing et al., 2017). For one data series, only one correction constant, the $(^{230}\text{Th}/^{232}\text{Th})$ activity ratio of the contamination, can be added to the calculation. Hence, in case several factors need to be explored, the data series requires repeated treatment.

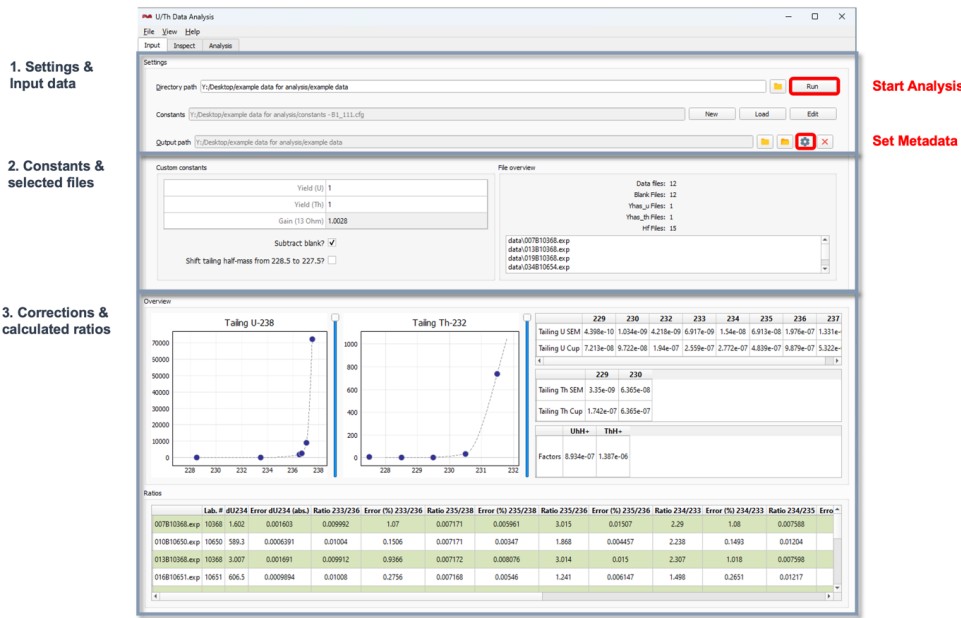

**Figure 1.** *Input tab: (1) In the top part the data folder is selected ("directory path"), the constants file ("constants") can be loaded ("load"), edited ("edit"), or created ("new"). In addition, it is possible to set an "output path". Red boxes show the "settings" button to enter metadata for saving, as well as the "run" button to start the analysis. Box (2) shows the custom constants box as well as the file overview for the selected folder. In box (3), the plots on the top left show the interpolated tailing. On the top right, numerical values of U and Th tailing and hydride correction are presented. The calculated ratios are shown in the bottom panel.*



Figure 1 shows the layout of the GUI 'Input' tab. Once the constants are implemented and the input data are
selected, it is optional to choose an output path to store the analysis output (Box 1). If no path is specified, the
results will be stored in the raw data folder. When clicking the settings button next to the output path (highlighted
in red in Box 1), a menu opens in which the following parameters about the sample can be noted: denomination,
type of archive, lab numbers, geographic origin, and a general description. The first and last laboratory number
are automatically read out from the raw data. The final output result files will then be saved in a newly created
folder under the name _[labnumber$_1$-labnumber$_n$] denomination in the directory chosen before. The metadata
information transferred through the GUI dialogue window is stored in a .json file in the respective folder. In the
'custom constants' panel (Box 2), some settings can be selected, for example, if the blank has already been
subtracted in the mass spectrometric software or not. Next to this panel, an overview over the files read in from
the folder is shown. After running the evaluation script with the loaded data and adjusted settings (Button "run",
highlighted in red in Box 1), the results of tailing and hydride correction, respectively as well as the calculated
ratios are displayed in the tab (Box 3). In addition, four excel .xlsx output files are created by the software at this
stage and stored in the directory path folder: Ratios.xlsx, Tailing.xlsx, PrBlank.xlsx and Intensities.xlsx.
Ratios.xlsx contains all calculated ratios and their errors as also presented in the GUI (10). Tailing.xlsx
summarizes the U and Th tailing values (in cps/V $^{238}$U) for each mass. In PrBlank.xlsx, the average values for
each mass of the procedural blank measurements before each standard and sample are presented. Intensities.xlsx
contains the full data tables, with the signals in cps or V for each mass over all cycles. Every standard or sample
has its own sheet.
The algorithm of the 'Input' tab starts by reading in the '.exp' measurement files for sample and standard
measurements, process blank (=instrumental background) and Uranium and Thorium abundance sensitivity
measurements. The lines for all cycles for all isotopes are imported into a pandas data frame. Firstly, matrices for
tailing, hydride and process blank correction are produced that are later subtracted from the isotopic masses used
for ratio building. The individual steps are carried out as follows:
- Tailing: Uranium tailing is determined by measuring the off-masses 228.5, 233.5, 236.5, 236.7, 237.05
and 237.5 before a measurement sequence starts. The first half-mass can be changed between 228.5 and
227.5 as we observed a scatter peak around this mass that switched its exact position every few months.
Tailing off-masses are 227.5, 228.5, 229.5, 230.5 and 231.5. For interpolation to full masses, we use
piecewise cubic Hermite interpolating polynomial fits (Kerber et al. 2023). The masses that undergo $^{238}$U
tailing correction are $^{233}$U, $^{234}$U, $^{235}$U, $^{236}$U, $^{229}$Th, $^{230}$Th and $^{232}$Th, while $^{232}$Th correction is applied to
$^{229}$Th and $^{230}$Th.
- Hydride isobaric interference: Hydride correction is determined by measuring 239 amu for UH$^+$ and 233
amu for ThH$^+$ during the abundance sensitivity measurements. The instrumental background (or
memory) is here referred to as process blank. It is measured between all sample and standard
measurements for 70 s. Typical blank levels afterwards are 0.5 cps for $^{230}$Th and 6 cps for $^{234}$U. The
matrices from these three corrections are then used for data reduction of each isotope.
- Detector setting: Three main different detector layouts are possible and are detected automatically by the
software: 1) all isotopes on cup, 2) $^{234}$U, $^{230}$Th and $^{229}$Th on SEM and 3) $^{234}$U on FC, $^{230}$Th and $^{229}$Th on
SEM. In normal operation, option 2) and 3) are used, depending on the $^{234}$U concentration of the





respective samples. $^{234}$U signals above 2 mV are measured on the centre FC which is the case for the
absolute majority of samples.
The treated data are now used for the calculation of all relevant isotopic ratios followed by subsequent outlier
tests, as described in Section 3.2.
**3.2     Inspect Tab**
Following the initial raw data treatment in the Input Tab, the 'Inspect' tab (presented in Figure 2) allows to
visualize and retreat the data prior to final age calculation. In particular, the settings for the outlier test can be
adapted.
The Inspect tab allows the user to plot the signal datapoints over the measurement cycle number for all isotopes
in the individual measurement files of the sequence. In the top of the tab (1), the ratio results table from the 'Input'
tab is presented. On the left (2), the list of measurement files (.exp) is shown. By clicking on a specific file, the
metadata and the signal plotted over the measurement cycle number are presented (3). On the bottom left (4), four
dropdown menus are available: The first one, "Isotope", allows to select one isotope from all of the isotope species
measured. "Mean" offers to switch between mean and median of the signal. The "Deviation" menu provides three
options for the assessment of data dispersion: standard deviation, median absolute deviation and interquartile
range. By setting "Scaling" to absolute or relative, the y-axis of the plot on the right can be changed between
signal intensities in V or cps and relative values. Any selection in the dropdown menus leads to an automatic
update of the plot on the right. Mean resp. median, as well as the dispersion ranges are presented as blue dashed
lines. Data points outside of the dispersion range are marked in red as outliers.

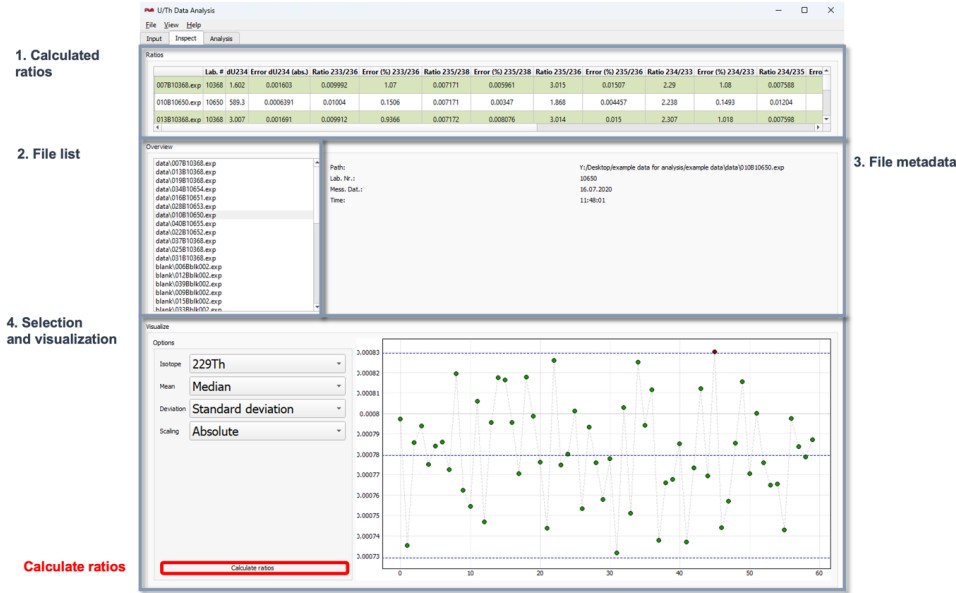


***Figure 2:*** *Inspect tab. (1) Ratio results table (from Input tab), (2) Overview of measurement files in folder, (3) metadata of a*
*selected file and signal over measurement cycle number for one isotope (which can be selected in (4)), (4) option selection*
*panel for the signal plotting.*



The "Calculate ratios" button (5) provides the option to recalculate the ratios using the updated mean and deviation
selection for all isotopes. The default settings are median and standard deviation. However, these updated options
are then used to exclude outliers from the ratio arrays, not the signal intensity arrays themselves. This means that
not necessarily exactly the same data points are marked as outliers in the signal intensity plots and will be
excluded, but the ones where signal ratios of two isotopes are outside of the accepted deviation range. The option
selected in "Mean" will then also be used to calculate the average of the isotope ratios.
The method of calculating the uncertainty of outlier-corrected isotopic ratios, however, is fixed:
$$err_{ratio} \ = \ 2 \, \frac{s}{\sqrt{n}} \tag{1}$$
with s being the standard deviation and n the number of data points (after outlier correction). The arithmetic mean
($\bar{x} \ = \ \frac{x_1 + \dots + x_n}{n}$) and the median $\bar{M}$ (central value of all values) are different ways of determining the average of a
distribution. The three different options for dispersion are defined as follows:
The standard deviation s is defined as
$$s \ = \ \sqrt{\frac{1}{n-1} \sum_{i=1}^{n} (x_i - \bar{x})^2} \tag{2}$$
with $\bar{x}$ being the mean. The interquartile range in turn is defined as the range containing the "middle" 50 % of
data points (Tukey, 1977). The median absolute deviation MAD is the median of absolute deviations from the
median, expressed as:
$$MAD \ = \ k_i \, M_i (|x_i - \bar{M}|) \tag{3}$$
with M being the medians and $x_i$ the original data (Leys et al., 2013;Huber, 2004;Rousseeuw and Croux, 1993).
For the calculation of the uncertainty MAD we assume normal distributed data, thus k =1.4286.

## 3.3   Analysis tab

In a last step, age calculation is carried out in the 'Analysis' tab presented in Figure 3. Here, additional input data
is necessary from the sample weight tables (1). There are several ways to import these tables: Either by clicking
"Load" and navigating to the respective folder, or by manually creating the table directly in the GUI ("Create").
An exemplary weight table is provided in the supplementary data (Figure 1). In the panel "Metadata history", the
previously loaded sample weight tables in the directory path folder are shown, and can be directly imported (2).
"Start Analysis" calculates the ages (3). Outputs are both presented in the GUI (4) and stored in an Results.xlsx
file. In case an output path was specified, Results.xlsx is created both in the output and in the directory path folder.
If the output path is missing, the file is only saved in the directory path folder. If an output directory has been
created for specific lab numbers, all following analysis of these same files will be written to the same output
directory, but not overwrite earlier Results.xlsx. The Results.xlsx has five sheets: *Inputs, Calc, Results, Constants*
and *Options*. *Inputs* presents sample weight and metadata as well as the calculated ratios. In *Calc*, all steps of the
age calculation such as concentrations and activity ratios are shown. *Results* is a summary of the most important
calculation steps and final age values and the same table as is presented in the GUI as (4). *Constants* contains the
whole list of values from the (potentially edited) '.cfg' file. In *Options* the average and dispersion measure option
are stored.



The equations for activity ratios to calculate ages are implemented according to Ivanovich and Harmon (1992),
with:

$$\left(\frac{234_U}{238_U}\right)(t) = \left(\left(\frac{234_U}{238_U}\right)_{init} - 1\right) \cdot e^{-\lambda_{234} \cdot t} + 1 \tag{4}$$

$$\left(\frac{230_{Th}}{238_U}\right) = 1 - e^{-\lambda_{230}t} + \frac{\delta^{234}U}{1000} \cdot \left(\frac{\lambda_{230}}{\lambda_{230}-\lambda_{234}}\right) \cdot \left(1 - e^{-(\lambda_{230}-\lambda_{234})t}\right) \tag{5}$$

with

$$\delta^{234}U = \left(\left(\frac{234_U}{238_U}\right)_{meas} - 1\right) \cdot 1000\ (\permil) \tag{6}$$

To obtain ages corrected for initial/detrital $^{230}$Th, the $^{230}$Th/$^{238}$U activity ratio used in eq. 5 is corrected using the
initial $(^{230}$Th/$^{232}$Th$)_{ini/detr}$ ratio and

$$\left(\frac{230_{Th}}{238_U}\right)_{corr} = \left(\frac{230_{Th}}{238_U}\right)_{meas} - \left(\frac{232_{Th}}{238_U}\right)_{meas} \cdot \left(\frac{230_{Th}}{232_{Th}}\right)_{ini/detr} \left(\frac{\lambda_{230}}{\lambda_{230}-\lambda_{234}}\right) \cdot e^{-\lambda_{230} \cdot t} \tag{7}$$

These equations need to be solved numerically. For the determination of age uncertainty, the usual approach is to
repeat the numerical determination of the age for several thousand runs in a Monte-Carlo simulation while random
sampling the input ratios from a normal distribution with μ corresponding to the ratio's value and σ corresponding
to the uncertainty on this parameter.

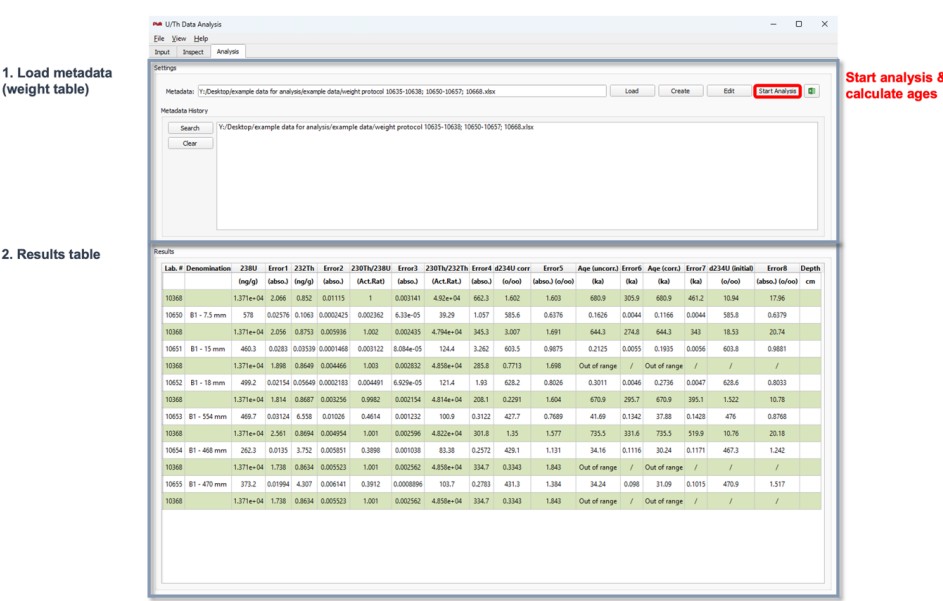


**Figure 3:** *Analysis tab. (1) Load sample weight tables (metadata files). The bottom panel lists the history of previously loaded*
*tables. The button highlighted in red starts the analysis to calculate ages ("Start Analysis") button. The panel in box (2)*
*displays the results table.*



## 4    Example dataset: Stalagmite B1

To demonstrate our data evaluation tool, we here present newly obtained ages of stalagmite B1 from Larga Cave, Puerto Rico. The results of activity ratios and calculated ages can be accessed in the supplementary table S1. Analysis of the speleothem samples reveals moderate U concentrations in the range between 300 and 600 ng/g, and minor detrital $^{232}Th$ contamination with ($^{230}Th/^{232}Th$) activity ratios of typically >300. However, in both the top 20 mm and around 450 mm dft lower ($^{230}Th/^{232}Th$) activity ratios of c. 40 – 125 are measured. U isotopic composition varies between 450 and 640 ‰ of $\delta^{234}U$ values. Uncertainties of the uncorrected ages are typically in the range of 0.2 to 0.6 % (Table S1). Drip water shows high U concentration of 0.825 ng/g and elevated initial Th concentrations, with an activity ratio of K=($^{230}Th/^{232}Th$)=11.1 ± 0.1. We have used the software to test how the chronology changes to assess the influence of a varying initial Th activity ratio. For this, we used three different correction models, including the measured initial Th ratio of the drip water (K=11.1 ± 0.1), the detrital correction value of K=0.75 ± 0.38 derived from the bulk Earth crust chemical composition, as well as a value of K=23.7 ± 7.5 as previously determined using isochrons on speleothem PR-LA-1 from the same cave (Warken et al. 2020). Figure 4 shows the ages corrected for initial $^{230}Th$ using the three different models. Only the initial $^{230}Th$ value measured in the drip water yields a stratigraphic order of the corrected ages supporting the use of this value. Residual variability around the mean chronology increases and age inversions appear in the record when using a different value of K.

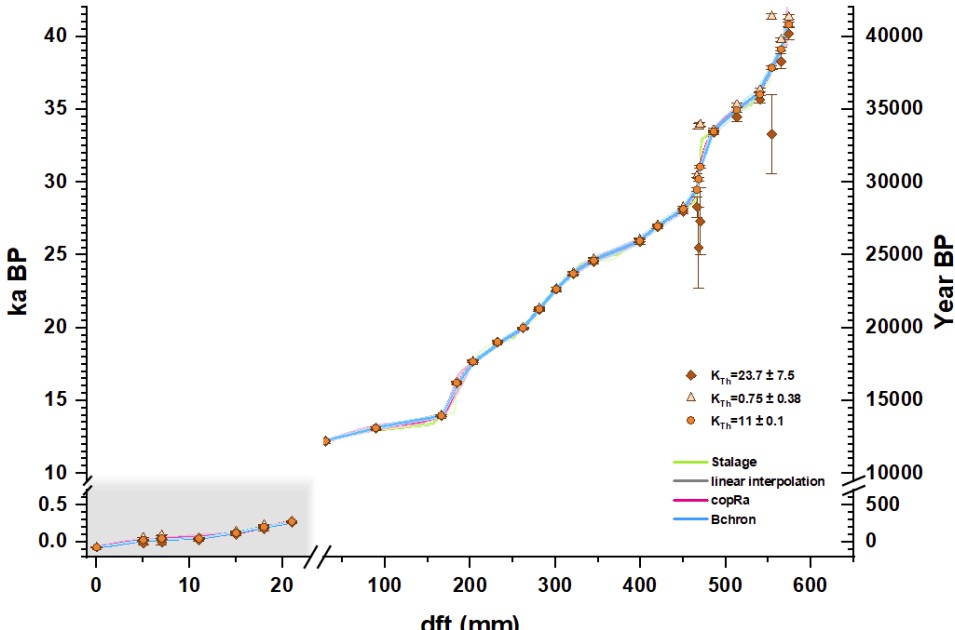

***Figure 4:*** *$^{230}Th/U$ ages and different age-depth simulations for stalagmite B1 using linear interpolation, as well as the algorithms Stalage, copRa and bchron linear interpolation, as well as the algorithms StalAge (Scholz and Hoffmann, 2011), CopRa (Breitenbach et al., 2012) and Bchron (Haslett and Parnell, 2008). Note that the axes are split at the position of the growth stop at 23mm dft to visualize the age-depth relationship also during the short growth phase during the latest Holocene after 0.3 ka BP.*



A Rosholt - Isochron determined for the section between 466 and 470 mm dft similar to the approach of Warken
et al. (2020) further supports the chosen correction value (supplementary Figure S3). This result highlights the
relevance to obtain such measures for better age correction either by studying the drip water $^{230}$Th and $^{232}$Th
isotope composition and using isochron approaches. The resulting corrected ages suggest a speleothem growth
between 0.060 ±0.013 ka BP$_{1950}$ and 40.81 ± 0.16 ka with a hiatus between 12.22 ± 0.043 and 0.277 ± 0.008 ka
BP at 23mm dft. Hence, speleothem B1 extends the existing speleothem record from Larga Cave into the
deglaciation covering Heinrich Stadial 1 (HS1), the so called Bølling/Allerød warming (BA), as well as the
beginning of the Younger Dryas cold reversal (YD).
Figure 6 shows the growth models obtained by linear interpolation, as well as the algorithms StalAge (Scholz and
Hoffmann, 2011), CopRa (Breitenbach et al., 2012) and Bchron (Haslett and Parnell, 2008) as implemented by
Roesch and Rehfeld (2020) (code accessed at https://github.com/paleovar/SISAL.AM, codes licensed by the right
holder(s) under a GPL-3). Growth rates of stalagmite PR-LA-B1 vary between c. 10 and 150 μm/a, with highest
values during the warm Bølling–Allerød period c. 13.97 ± 0.051 and 13.114 ± 0.073 ka BP as well as the late
Holocene growth phase after 0.277 ± 0.008 ka BP. Lowest growth rates occur during the final stage of Heinrich
Stadial (HS) 1 (16.23 ± 0.082 to 13.97 ± 0.051 ka BP), HS3 (31.02 ± 0.10 to 29.38 ± 0.12 ka BP), and HS4 (40.81
± 0.16 to 39.12 ± 0.12 ka BP).

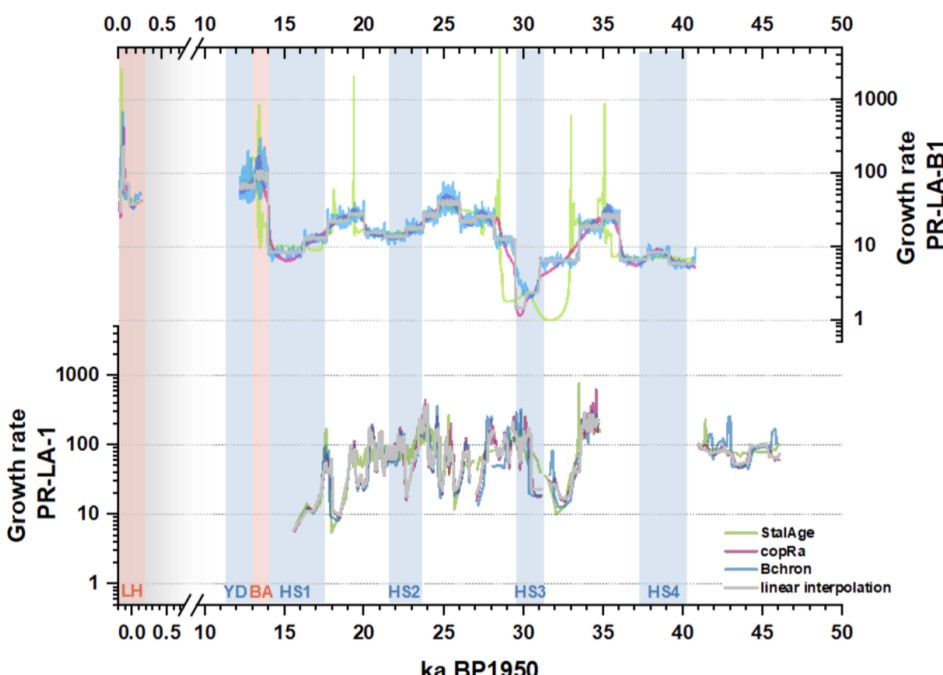


**Figure 5:** *Growth rates of speleothems B1 (top panel, this study) and PR-LA-1 (Warken et al., 2020). Vertical red (blue) bars*
*indicate the timing of warm (cold) phases in Puerto Rico, including the growth phase of B1 during the latest Holocene (0.3 ka*
*BP to present), the Younger Dryas (YD), Bølling/Allerød warming (BA), and Heinrich stadials (HS) 1 to 4.*



## 5 Discussion

### 5.1 Outlier correction

Outlier correction is carried out automatically by the software adapting the dispersion measure of the raw data and in the following we argue that generally means should be replaced by medians. Shao et al. (2019) had addressed this problem by implementing manual outlier removal by comparison to boxplots based on interquartile ranges. We opted for the automatic version as this is more time efficient for large datasets. The different dispersion measure options described in Section 3.2 are relevant because measurements are not always ideal cases with normally distributed data and thus outliers. During measurements, short-term system instabilities occur for a variety of reasons, such as varying gas flow in the inlet system, plasma instabilities, and varying size of sample aerosols causing outliers in the signal intensities. Even though only the ratios between the different isotopes are of interest, strong changes in signal intensity may lead to varying isotope ratios, as a result of changing variance. Such difference may be amplified by the use of different detectors or with respect to different magnetic field settings, which are not necessarily responding at exactly same amplitude. Moreover, signal decreases (detuning events and temporal clocking) cause the statistical variance to increase locally.

Figure 6 shows an example: The upper panel displays periodic dips in the $^{238}$U signal intensity during a measurement. In the lower panel of Fig. 4, the uncorrected ($^{230}$Th/$^{238}$U) activity ratio for the same measurement is plotted. For both curves, the different measures to calculate dispersion are shown. The default method (2 standard deviation) does not remove all the systematic outliers. Also, it is clearly visible that the median agrees much better with the majority of signal intensity values than the mean, which is much stronger influenced by the periodic dips due to the asymmetry in the statistical distribution. Such an obvious difference is not visible in the isotope ratio, but within the resulting uncertainty. Consequently, we propose to generally use the median instead of mean by default. This is more accurate in the case of asymmetric small-scale oscillations inside the non-outlier interval and has no disadvantages.

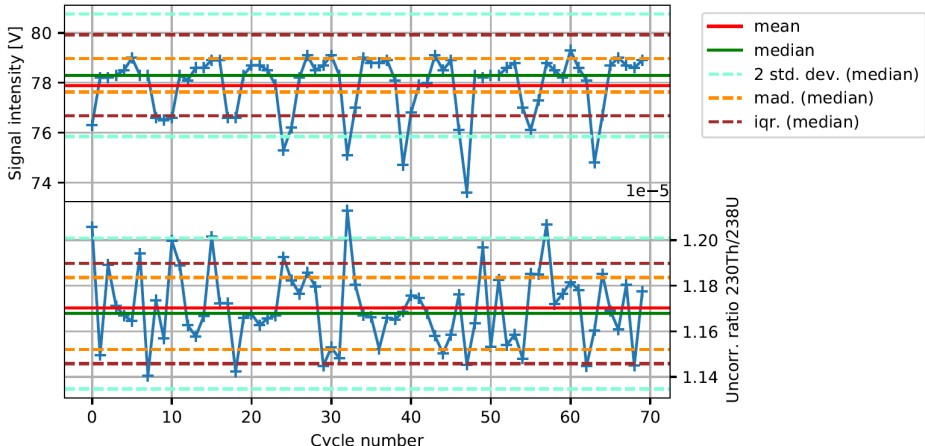

***Figure 6:*** *Upper panel: $^{238}$U signal intensities in Volt over measurement cycles for a carbonate sample during routine lab measurements. Lower panel: Corresponding uncorrected $^{230}$Th/$^{238}$U ratio. Mean and median as well as the three different dispersion measures are plotted.*



Applying standard deviation as a dispersion measure in Figure 6 does not cover most of these outliers due to their
large number and relatively small deviation. Thus, applying another dispersion measure for outlier removal is
necessary here, and in addition more robust and easier to accomplish than manual deletion of all of the outliers. It
is important, however, to stress that the outlier correction using the selected dispersion option is run on the
calculated ratios after correction, not on the signal intensities themselves. This implies that when all isotopes are
affected in the same way, they pass the outlier test. This is, however, unlikely at least for ratios of isotopes
measured in different magnetic field settings. The dispersion measure of the outlier corrected ratio array is the
same in every case, as described by Equation 1.

**5.2 Detrital thorium correction**

Thorium correction is often crucial for studying carbonates where the correction is significant, but the initial $^{230}$Th
value is unknown, potentially variable, or when studying "dirty" carbonates such as tufa and travertine (Mallick
and Frank, 2002;Hellstrom, 2006;Wenz et al., 2016). Several studies have shown that this correction particularly
important for speleothem records from the Caribbean and Central American region, where values where found
including 2 ± 1 (Schorndorf et al., 2023) or 14 ± 4 (Moseley et al., 2015). In Larga Cave, initial ($^{230}$Th/$^{232}$Th)
ratios are presumably even higher (Vieten et al., 2024;Warken et al., 2020). Besides the terrestrial regime, this
aspect is also relevant for marine archive such as corals, where studies propose a large range of seawater
($^{230}$Th/$^{232}$Th) activity ratios. While Cheng et al. (2000a) set the range to 80 ± 80 for deep-sea solitary corals, and
Frank et al. (2004) calculated 10 ± 4 from seawater in the Eastern North Atlantic deep sea, values between 0.4 –
3.1 were determined for tropical corals (Shen et al., 2008). The range of both absolute values and uncertainties
for these widely studied archives is hence enormous, and the choice of the appropriate correction model becomes
particularly important, when (i) samples are very young and have generated only small amounts of $^{230}$Th from U-
decay, or (ii) when ultra-high precision is at play since any possible correction of the data contribute to the final
age uncertainty. In our case study, we have run the correction of the ages of stalagmite B1 using three different
correction models (Table S1). The resulting differences are visualized in Figure 4, and demonstrate the significant
impact not only on absolute corrected ages, but also their uncertainties. For the young age at 7 mm dft
(0.0466±0.0045 ka BP), the difference in the absolute corrected age when using another correction factor than the
drip water value of K = 11.1±0.1 is c. ±50 years, which corresponds to a relative difference in the order of 100%
(compare Table S1). Another example is the sample at 554 mm dft (37.81±0.14 ka BP for K=11.1±0.1), for which
the other correction models also lead to substantially different ages of 41.37±0.19 ka BP (K=0.75±0.38) and
33.3±2.7 ka BP (K=23.7±7.5), hence the differences are still in the range of c. 10%. Notably, the low relative
error of the initial ($^{230}$Th/$^{232}$Th) activity ratio of the drip water results in equally low uncertainties of the corrected
age in the range of 0.4%. In contrast, the relative uncertainty of the age corrected with K=23.7±7.5 increases to
8%. Our GUI permits an easy adjustment of the initial ($^{230}$Th/$^{232}$Th) activity ratio for Th correction, which allows
a direct assessment of the resulting corrected ages and uncertainties, and provides thus a convenient basis for
further comparisons of the data. The use of a standardized software instead of handmade tuning reduces the
susceptibility to potential errors, e.g., from copy-pasting, and ensures reproducibility in case a re-evaluation of
the data is required to a later stage.



### 5.3 In-cave comparison of speleothem growth rates

The high number and precision of $^{230}$Th/U ages of speleothem B1 allows investigation of growth rates changes.
Comparison with northern hemispheric climatic changes suggests, that speleothem B1 growth is sensitive to
prominent millennial-scale temperature variability, with higher growth rate during warmer phases and vice versa.
In particular, during the cooler and drier Heinrich stadials (Warken et al., 2022b), growth rates are reduced.
In addition, the results allow a comparison of the two coeval stalagmites from Larga Cave as shown in Figure 5.
Overall, GRs of PR-LA-B1 are about 5 times lower than observed for PR-LA-1, where average annual growth
rates are up to several mm/a. The difference in mean GR is also reflected in the shape of both speleothems, with
PR-LA-1 exhibiting a large and variable diameter between c. 15 and 35 cm (Warken et al., 2020), while B1 is
thinner with a diameter of 10-15 cm (supplementary Figure S1B). Differences in speleothem growth rates and the
shape of a stalagmite may result from temperature, carbonate saturation, drip rate, and carbon dioxide contrast
between cave air and saturation concentration of drip water (Merz et al., 2022;Skiba and Fohlmeister,
2023;Kaufmann, 2003;Dreybrodt, 1999). Ca concentrations in Larga Cave show no significant differences
between drip sites (Vieten et al., 2018a, Vieten et al., 2018b, Warken et al., 2022), and a single Ca concentration
measurement at site B1 is also within the same range (Vieten et al., 2018a, Vieten et al., 2018b).
Therefore, the amplified GR and generally larger diameter of PR-LA-1 could be the result of the considerably
lower pCO2 values in the main passage (600 and 1800 ppm) than compared to the back part of the cave (2300
and 3600 ppm), which facilitates enhanced oversaturation of the drip water with respect to calcite, and hence,
stronger degassing of $CO_2$ and speleothem growth (Merz et al., 2022). In addition, variations in drip rates influence
the GR. Spot observations suggest a multi-annual variability of drip rates at the location of stalagmite PR-LA-B1.
For PR-LA-1, no modern drip site is available, which precludes direct comparison of the role of the drip rate. The
factor five amplified GR of specimen PR-LA-1, however, gives rise to presume a higher drip rate in addition to
lower atmospheric cave air pCO2 values, in particular during centennial increases of the GR to values as high as
mm/a at this site. Further note that the timing of the GR increases above the mean GR or paused growth did not
occur synchronously, which precludes a common mechanism stimulating the GR. Hence, the two stalagmites
reveal growth differences potentially related to ventilation conditions.
Other differences of the two sites are visible in both speleothems geochemistry, which, however, cannot be directly
related to drip rate or cave air pCO2 concentration. The Uranium concentration [U] of PR-LA-B1 is systematically
higher than for the one of PR-LA-1 (c. 90 – 400 ng/g) and the initial $\delta^{234}$U is with values ranging between 450 to
640 ‰ strongly elevated compared to the $\delta^{234}$U from PR-LA-1, which varies between values of c. 70 - 100 ‰.
This demonstrates a reduced flux of excess $^{234}$U from the host rock at the drip site of PR-LA-1, potentially resulting
from varying release of $^{234}$U through alpha-recoil of the decay of $^{238}$U at the two sites. A likely explanation may
be the difference in local host rock overburden of PR-LA-B1 with c. 40-60 m to PR-LA-1 with c. 20-40 m, and
thus moderately longer residence times of the karst water at site B1. Consequently, given the sum of observations
it seems most likely that the GR of PR-LA-1 in the better ventilated region with less rock overburden responds to
drip rate more sensitively than PR-LA-B1, which in contrast seems more sensitive to cave ventilation, i.e., cave
air pCO$_2$.



## 6 Conclusion

We here provide an algorithm combined with a user-friendly GUI application for $^{230}$Th/U MC-ICP-MS data treatment and age calculation. The two so far published programs explicitly aimed at $^{230}$Th/U dating data reduction and age calculation are both written for ThermoFisher Neptune instruments as well. Pourmand et al. (2014) described a Mathematica routine, distributed as a Computable Document Format (.cdf) file, while Shao et al. (2019) had published a Matlab algorithm with GUI. We here have chosen to use Python for our algorithm and GUI to keep it open-source. The advanced user might want to change settings, which makes an opensource language and libraries a major advantage. However, the stand-alone executable .exe format of the GUI allows user-friendly handling also for non-programming experts. Our program supports multiple types of detector configurations: the FC-FC based approach as well as FC-SEM combining protocols. It is however adapted for combined Th and U measurements in three magnetic field lines (compare Kerber et al. (2023)), but other methods (such as separate solutions for Th and U) can be implemented with small changes in the code. Furthermore, we offer the first order Taylor derivation as a time-saving option for uncertainty calculation of final ages. Our application is especially designed to take reproducible and clear data management into account by a collection of methods: This includes that automatic creation of folders containing the results files and information on the sample metadata is possible and that .xlsx output files automatically contain all constants used for calculation, as well as the settings for outlier correction. Manually changing input constants, e. g. correction, of initial/detrital Th does not require to go to the code directly. So, the whole analysis scheme does not require any copy-and-pasting from one excel table to the other, and the constants used for calculation are easy to update.

Lastly, we demonstrated our protocols and data analysis scheme by accurately measuring and evaluating 30 speleothem ages from Larga Cave, Puerto Rico. Analyses of the growth rates and comparison with a coevally growing stalagmite from the same cave highlights the importance of in-cave processes for speleothem deposition rates.

## Author contributions

IK - conceptualized the work, created and tested the implementation and operation of the code, co - supervised FK, who developed the code for the GUI and tested rigorously all corrections. NF - conceptualized the project, supervised IK, and FK and quality controlled the Th U isotope measurements of PR-LA-B1. SW - conceptualized the project, provided guidance on sample selection, verified the code and conceptualized the application. SW further evaluated the resulting age data on PR-LA-B1 and supervised a student project during which these and other data had been collected.

## Code availability

The source code of "UTh Data Analysis" is accessible at  https://github.com/EnvArchivesHD/UTh_Analysis . The folder https://github.com/EnvArchivesHD/UTh_Analysis/tree/main/dist contains the compiled .exe file for the GUI ("UTh Data Analysis.exe") as well as default configuration files ("constants – coral.cfg" and "constants – stalag.cfg".



**Data availability**

Results of speleothem B1 $^{230}$Th/U dating are available in the online supplementary material.

**Sample availability**

Sample material is available on request to swarken@iup.uni-heidelberg.de

**Competing interests**

At least one of the (co-)authors is a member of the editorial board of Geochronology.

**Disclaimer**

**Acknowledgements**

The authors are very thankful to the enormous support of the whole team of the research group „Physics of Environmental Archives" at Heidelberg University. Special thanks go to R. Eichstädter and A. Schröder-Ritzrau for continuous engagement in the laboratory work and quality control. J. Arps is thanked for the development of a previous version of „UTh-Analysis". We are particularly grateful to R. Vieten for continuous support of speleothem research in Larga Cave. R. Vieten, N. Schorndorf, S. Therre and J. Förstel are thanked for their help in the field and with sample collection. We greatly acknowledge the work of N. Schorndorf, J. Schandl, J. Gafriller, and A. Mielke on the chronology of speleothem B1. J. Bühler, C. Roesch, and K. Rehfeld are thanked for providing access and support with the age-depth modelling code. N. Frank received financial support for $^{230}$Th/U measurements (DFG Grant N°256561558) and for the installation of the MC-ICPMS facility (DFG Grant N°247825108). S. Warken received financial support for the climate study of Puerto Rican speleothems via the DFG (Grant N° 512385350) and by Heidelberg University via the Olympia Morata program.

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
