# Peer review of "Technical note: "U-Th Analysis" – an open-source software dedicated to MCICPMS U-series-data treatment and evaluation"

_EGUsphere, 2024_

## Author Comment (AC1)

We greatly appreciate the helpful and constructive comments and suggestions. In the following find our detailed Response

**Response to Reviewer #2**

This paper presents a Python-based program "U-Th Analysis" designed for U-Th isotopic data analysis for MC-ICPMS U-series dating. The program has a friendly graphical user interface and is open for the source code, which can be modified according to the measurement methods. This program comprises many automatic functions: e.g., to calculate U and Th tailing effects, identify outliers in isotopic ratios, correct for initial 230Th contamination, as well as to load and save data. I support to accept this paper for publication, as a transparent data treatment algorithms is desired for each of dating laboratories.

I have a few comments:

U-Th isotopes can be measured using MC-ICPMS with different methods, and in many labs, the two elements were measured separately. In the Conclusion of this study, it was mentioned that the U-Th were measured simultaneously. I think the isotopic measurement method should be clarified before describing the program, probably in the section of "2.2. Chemical preparation and instrumentation".

Many thanks for this comment, we will add a section expanding the method description in section 2.2, e.g.:

"The cup setting to collect isotope signals on masses 238 to 229 is shown in table 2 in Kerber et al. (2023). The first cycle collects all U isotopes for 2 seconds, with $^{234}$U on the central detector (FC/SEM). The second and third cycle collect the Th isotopes for 2 seconds integrations time, with $^{230}$Th and $^{229}$Th on the central SEM. These cycles are repeated for an optimal number for each measurement."

Since reviewer #1 also commented similarly, we will add clarifying statements that the GUI in its present form is optimized for our setup, but can in principle be adapted for other protocols and instrumentation.

If I understand well, the "Input" tab performs corrections of blank, tailing, and hydride, the "Inspect" tab filters the outliers from isotopic ratios, and the "Analysis" tab calculates the U-series age and corrects for initial 230Th contaminations. The combination of FC and SEM was used for U-Th measurements, so, I want to know how the SEM/FC yield is corrected, and how mass fractionation is corrected, and if the uncertainties of the two factors are considered in the program.

Many thanks again for this comment and we agree that this information has been missing. We suggest to add the following statement to section 2.2:

"Mass fractionation (or mass bias) is corrected via the natural ratio of $^{235}U/^{238}U$ due to the lack of natural Th isotopes. In our setup, the ratio of the artificial isotopes $^{233}U/^{236}U$ ratio in the spike is monitored for double checking the mass bias correction. The calibration of FC gain and SEM yield is described in detail in Kerber et al., (2023): While there is an internal electronic calibration function for the calibration of $10^{10}$ Ω and $10^{11}$ Ω amplifiers, the $10^{13}$ Ω amplifier in our setup was in this study calibrated manually. For this, the gain factor was determined regularly by measuring $^{235}U$ alternately on the $10^{13}$ Ω and $10^{11}$ Ω amplified cup. In an analogous manner, the SEM yield is routinely determined by measuring $^{235}U$ alternating on the SEM and on a $10^{11}$ Ω FC at a signal intensity of $\sim$ 5 mV. Since HU-1 standards are measured with the same detector configuration in standard bracketing mode, the observation of the ($^{234}U/^{238}U$) values of HU-1 measurements allows monitoring and manual optimization of gain and yield values for each measurement sequence in the data analysis scheme."

In line 211, it was mentioned that the instrument background is measured between all sample and standard measurements for 70 s. Is the background measured on SEM by peak-jumping or by the combination of SEM and FC? For the blank correction, are the outliers in the blank measurements filtered.

We thank for this comment. The instrumental background is measured by the combined protocol of SEM and FC. The same outlier correction is applied for both sample and blank measurements and is adjusted accordingly when changes are made in the "inspect tab".

We will add a clarifying statement to the methods description:

"Samples, standard, and procedural blanks are measured with the same configuration."

A few typos:

Line 30: The "manifold in geochemistry, archaeology, and climate science" was repeated.

We will change this accordingly.

Line 192: What is the "GUI (10)"?

We will change this leftover from a former version of Figure 3.

Line 205: It should be "Thorium tailing".

We will change this accordingly.

Line 323; It should be "Figure 5".

We will change this accordingly.

---

## Author Comment (AC2)

We greatly appreciate the general assessment of our open source software and manuscript by reviewer 1. In the following find our detailed Response to the specific comments

**Response to Reviewer #1**

SPECIFIC COMMENTS

It is not explicitly stated that the GUI is for handling data from ThermoFisher Neptune instruments only or how data from other instruments can be handled. This would be beneficial to users of the 230Th/U dating method with data from other instruments.

We thank Reviewer #1 for this hint. We will add a clear statement in the publication. In the current state, the GUI is handling only data from ThermoFisher Neptune instruments. However, the Python-based code should be easily adapted for other instruments data formats.
We suggest, e.g., to add the following statement to the last paragraph of the introduction:

"In this study, we present our user-friendly GUI and the underlying algorithm for data treatment and age calculation. The software is currently developed and optimized for ThermoFisher Neptune MC-ICP-MS instruments, but the open-source code in principle allows adaptations to other setups, instruments, and data outputs."

We will also change the first sentence of the conclusions to:

"We here provide an algorithm combined with a user-friendly GUI application for the treatment of 230Th/U MC-ICP-MS data obtained by ThermoFisher Neptune instruments, and subsequent age calculation and correction."

TECHNICAL CORRECTIONS

Line 51: The use of "(i)" is confusing since no list is provided.

We will remove this.

Line 62: "initial Thorium" should be "initial Th" for consistency within the same sentence and entire document.

We will change this.

Line 259: In this section "3.3 Analysis tab" the numbers in parentheses are not consistent with Figure 3. For example, line 265 makes reference to (3) for "Start Analysis" and (4) for presentation of the results, but there is no (3) and (4) in Figure 3.

We apologize for the inconvenience; we will change the numbers in the text.

Line 263: The exemplary Excel file for the weight table provided in the supplementary data should have its sheet names and column headers translated from Deutsche to English for accessibility.

We will change this file and upload a translated version.

Line 310: Delete the first "as well as the algorithms Stalage, copRa and bchron linear interpolation"

We will delete this duplication.

Line 316: Caption for supplementary Figure S3 has "223Th" instead of "232Th"

We will change this accordingly.

Line 353: "which is much stronger influenced by" please revise. Strongly instead?

We will change to "strongly influenced by".

Line 373: "this correction particularly" is missing "is"

We will insert "is".

---

## Author Response (AR1)

Comments of the editor

The two reviewers have only minor comments on your Technical Note, which you have adequately addressed in your response letters. I am therefore happy to recommend publication of your paper in Geochronology pending these changes.

Dear editor, we thank for the recommendation as well as the additional comments. Please find our responses in the following. Please note that some of the requests have required testing the software and changing the code. Due to this work we have now added Aaron Mielke as a coauthor, because he found all the code mismatches and will in the future further develop this application. He is also an expert in Th/U dating and has helped revise the manuscript. Thus, we prefer to have him as a coauthor, rather than just thanking him for his help in the acknowledgements.

However, I would also like you to address a few additional points which I have found after reading the paper myself:

1. Lines 29 and 30 contain duplicate sentences and use the word "manifold" instead of "manyfold".

Changed.

2. Equations 1 -3 are unnecessary. Standard deviation, standard error, interquartile range and median absolute deviation are commonly used terms that do not need explicit definitions in a paper of this kind.

We have shortened this section and deleted equations 1-3. The paragraph now reads as follows:

"In total, the software provides three different options for dispersion, including (i) the standard deviation (s), (ii) the interquartile range (IQR) (Tukey, 1977), and (iii) the median absolute deviation (MAD) (Leys et al., 2013; Huber, 2004; Rousseeuw and Croux, 1993). For the calculation of the MAD we assume normal distributed data."

3. Line 245 states that your software uses means of ratios. This treatment has some undesirable statistical properties (as detailed by, for example, Ogliore, NIM-B, 2011 and McLean et al., G-Cubed, 2016). It would be useful to offer your users the ratio of the means as an alternative solution.

We agree with the editor that the ratio of the means may bias the results under very low count rates or fast changing signals, which is generally in optimized liquid MC-ICPMS analysis not the case. We propose to implement a further note and treatment of this aspect in the next version of the software. The text now reads in L276: The treatment of means of ratios may have undesirable statistical properties for low or fast changing signals (Ogliore, NIM-B, 2011, McLean et al. 2016), which could be taken into consideration when updating the software.

4. Line 297: What does "dft" stand for?

*"dft" stands for distance from top. We have included the definition of this abbreviation now.*

5. Sections 4 and 5.3 can be shortened. People will read your technical note to learn more about your software. Most of them probably won't care so much about its application to Speleothem B1. As far as I can tell, Figures 4 and 5 were not generated by your software. I think they can be moved to the supplement.

*We thank for the comment and have shortened section 4 and 5.3 as suggested. For example, we have moved both Figure 4 and 5 to the supplement as new Figures S3 and S4, and removed Figure S4 and the associated discussion of the isochron.*

6. Line 338 suggests that means should be replaced by medians. However, Section 3.3 says that the software uses means (of ratios). Which is correct?

*As outlined in section 3.2, the user has the choice to use either the mean or the median. This is possible in the "inspect" tab in the drop-down menu "mean". The default setting of our software is the median.*

7. Line 339 states that outlier rejection is done using box plots. Can you clarify how this works? The convention is that box plots define outliers as being more than 1.5 IQR above or below the median. Did you follow this definition?

*We assume the reviewer refers to the sentence "Shao et al. (2019) had addressed this problem by implementing manual outlier removal by comparison to boxplots based on interquartile ranges. We opted for the automatic version as this is more time efficient for large datasets.". Hence, we are not using box plots for outlier definition. For details of the software of Shao et al., we refer to their publication.*

8. According to line 351: "The default method (2 standard deviation) does not remove all the systematic outliers". What do you mean with "default method"? A 2 sigma cutoff is a very poor outlier detection criterion, as it rejects 5% non-outliers. This is called a "type-1" error in statistics. If your software includes a 2-sigma criterion, then please remove it.

*For the calculation of the mean ratios, different mean and dispersion measures are available. We have removed this confusing sentence the editor refers to in this paragraph. The (dis-)advantages of different dispersion measures are discussed in more detail the following paragraph.*

9. Line 384: Supplementary table S1 uses three widely different initial $230Th/232Th$ ratios (0.75, 11 and 75). One of these ($[230Th/232Th] = 11$) is based on drip waters. How were the other estimates obtained. Lines 62-71 mention isochrons and independent age constraints as alternative means of estimating the detrital component. Did you use those? Does your software perform isochron regression? If so, does it implement the Ludwig and Titterington (1994) algorithm. Some more details would be useful here.

*Our software does not perform isochron regression. To prevent confusion, we have removed the isochron calculated for stalagmite PR-LA-B1 in the supplementary material and the related discussion.*

The different mentioned ($^{230}$Th/$^{232}$Th) correction models are based on different previous constraints. The value of 0.75 follows the conventional approach assuming a upper continental crust $^{232}$Th/$^{238}$U weight ratio of 3.8 (Taylor & Mvlennan, 1985) with an uncertainty of 50% (Ludwig & Paces, 2002) and $^{230}$Th, $^{234}$U, and $^{238}$U in secular equilibrium for the detrital material to account for initial Th. The value of 23.7 stems from a previous analysis of Warken et al. (2020), who constrained the initial Th ratio by using an isochron approach on a speleothem from Larga Cave. We have added the relevant citations and references to the supplementary table S1, and have updated the explanations in the main text accordingly.

10. Although the Windows executable is useful, it would also be helpful if your code would work on other operating systems as well. However, when running your Python code on my computer (Ubuntu 22.04), I get the following error message:

Traceback (most recent call last):
File "/home/pvermees/temp/UTh_Analysis/main.py", line 259, in
GUI = Window()
File "/home/pvermees/temp/UTh_Analysis/main.py", line 40, in __init__
self.inputTab = InputTabWidget(self, self.ratioBuilder)
File "/home/pvermees/temp/UTh_Analysis/InputTabWidget.py", line 35, in __init__
self.initOverviewBox()
File "/home/pvermees/temp/UTh_Analysis/InputTabWidget.py", line 509, in
initOverviewBox
self.uTailTable.setVerticalScrollMode(QtGui.QAbstractItemView.ScrollPerPixel)
AttributeError: module 'pyqtgraph.Qt.QtGui' has no attribute 'QAbstractItemView'

We thank for this hint. We apologize for the inconvenience that the code did not run on your system. We have identified this error as arising from running the code on a different Python (and package) version. We have updated and tested the code and it should now run with the latest Python versions on different operating systems including MacOS, Windows and Linux.

Many thanks again for helping us to improve the manuscript and to make the code best available.

The authors

---

## Author Response (AR2)

Author's Response to minor review by Peter Veermesch

Comments

Public justification (visible to the public if the article is accepted and published): The response to the reviewer comments is satisfactory, although it is unfortunate that the authors postpone the implementation of alternatives to the mean of the ratios until a future release. I won't press on this point too hard, but I do hope that the authors keep their promise to maintain the code.

> Dear Pieter Vermeesch, thank you for your kindness to not push too hard. We have all information and papers viewed and are in the process of implementation of mean of the ratio, however, this has turned out somewhat more difficult technically than anticipated as we need to modify the input data arrays, which demands some extra work on the structure of the software. We apologize, for not directly responding, but it will be done as promised soon.

Importantly, I have now succeeded in running the software on my computer. However, I only managed to do so by deviating from the settings shown in Figure 1. There is no folder called "example data" in the "example data for analysis" folder of the supplementary data. There is a folder called "data", but the software crashes when I use this. I had better luck after selecting the top-level "example data for analysis" folder. I don't know if this changed behaviour is related to modifications made to the code after the manuscript. In any case, Figure 1 probably needs updating.

> We apologize for this inconvenience. In fact, you found the correct solution, and we have now changed the screen shot of Figure 1 accordingly. We have also further revised the numbering of figure 1 and 4 in two places. We sincerely hope the article is now ready for publication and thanks you ones again for the time and helpful comments that significantly improved the manuscript.

---

## Author Response (AR3)

Ruprecht-Karls-Universität Heidelberg
**Institut für Umweltphysik**

[Figure]

Institut für Umweltphysik · Neuenheimer Feld 229 · D-69120 Heidelberg

**Prof. Norbert Frank**

Im Neuenheimer Feld 229
69120 Heidelberg
Tel:  +49 (0)6221) 54 6332
Fax:  +49 (0)6221) 54 6405
Norbert.Frank@uni-heidelberg.de
http://www.iup.uni-heidelberg.de

**Datum:** 06.11.2024

EGUSPHERE
Editor: Pieter Vermeesch

Dear Pieter Vermeesch

Please find enclosed all the necessary files for the final production and publication of our article egusphere-2024-1788.
We greatly appreciate the excellent work of the journal team and reviewers and thank you very much once again for helping us to improve the quality of the manuscript and the open source software.

Sincerely yours

Prof. Norbert Frank